# Transfusion Transmissible Infections in Blood Donors in the Province of Bié, Angola, during a 15-Year Follow-Up, Imply the Need for Pathogen Reduction Technologies

**DOI:** 10.3390/pathogens10121633

**Published:** 2021-12-17

**Authors:** Luis Baião Peliganga, Vinicius Motta Mello, Paulo Sergio Fonseca de Sousa, Marco Aurelio Pereira Horta, Álvaro Domingos Soares, João Pedro da Silva Nunes, Miguel Nobrega, Lia Laura Lewis-Ximenez

**Affiliations:** 1Viral Hepatitis Laboratory, Oswaldo Cruz Institute, Oswaldo Cruz Foundation, Rio de Janeiro 21040-900, Brazil; peligangabaiao@aluno.fiocruz.br (L.B.P.); viniciusmello@aluno.fiocruz.br (V.M.M.); sousa@ioc.fiocruz.br (P.S.F.d.S.); 2Disease Control Department, National Directorate of Public Health, Ministry of Health, Luanda, Angola; 3Internal Medicine Investigation Department, Faculdade de Medicina da Universidade Agostinho Neto, Luanda, Angola; 4NB3 Laboratory, Oswaldo Cruz Institute, Oswaldo Cruz Foundation, Rio de Janeiro 21040-900, Brazil; marco.horta@fiocruz.br; 5Hospital Geral do Bié, Cuito, Angola; aldoso43@hotmail.com; 6Laboratory of Experimental Vaccines, Paulista School of Medicine, Federal University of São Paulo, São Paulo 04039-032, Brazil; jps.nunes07@unifesp.br; 7Independent Researcher, Brooklyn, NY 11201, USA; mtnobrega@gmail.com

**Keywords:** TTIs, blood donors, Angola

## Abstract

Transfusion transmissible infections (TTIs), caused by hepatitis B virus (HBV), human immunode-ficiency virus (HIV), hepatitis C virus (HCV), and syphilis, have a high global impact, especially in sub-Saharan Africa. We evaluated the trend of these infections over time in blood donors in Angola. A retrospective cross-sectional study was conducted among blood donors in Angola from 2005 to 2020. Additionally, frozen samples obtained from blood donors in 2007 were investigated to identify chronic HCV carriers and possible occult HBV infection (OBI). The overall prevalence of HBV, HCV, HIV, and syphilis was 8.5, 3, 2.1, and 4.4%, respectively, among 57,979 blood donors. HBV was predominant among male donors, while the remaining TTIs were predominant among women. Donors >50 years had a significantly high prevalence for all TTIs. Chronic HCV infection was ab-sent in 500 samples tested and OBI was present in 3%. Our results show the continued high prev-alence of TTIs among blood donors in Angola. Most infections showed a significantly low preva-lence in years with campaigns seeking voluntary blood donors, thus, reinforcing the importance of this type of donor to ensure safe blood. Africa, with a high prevalence of diverse pathogens, should consider cost-effective pathogen reduction technologies, once they are commercially accessible, to increase the availability of safe blood.

## 1. Introduction

Transfusion transmissible infections (TTIs) have a high global public health burden. An estimated 354 million individuals worldwide are chronically infected with hepatitis B virus (HBV) or hepatitis C virus (HCV) [1], which caused 1.1 million deaths globally in 2019 [1]. This number has surpassed that of deaths caused by the human immunodeficiency virus (HIV), which for 2020 was estimated at 680,000 globally. HIV infects approximately 37.7 million people worldwide, with 67% (25.4 million) living in Africa [1,2]. For syphilis, an estimated 18 million people were infected in 2012, representing 0.5% of the global population. Updated data for 2020 estimated 7.1 million new cases of syphilis worldwide, which surpasses the total of 4.5 million cases for the other three TTIs combined [1].

To guarantee safe blood for transfusion and to prevent TTIs, candidate blood donors should be screened for HBV, HCV, HIV, and syphilis infections [3,4]. Blood transfusion is an essential life-saving therapy in Africa where severe malaria, post-partum haemorrhages, malnutrition, and sickle-cell anaemia are highly prevalent [5]. Nevertheless, blood transfusion remains a challenging process in developing countries, particularly Sub-Saharan Africa (SSA) [5,6]. This concern arises primarily because of the irregular supply of test kits, which is one of the most reported barriers to screening, in addition to insufficient financial resources [5,6,7]. Furthermore, this region has a high prevalence of TTIs: HBV (6.1%), HCV (1.6%), HIV (0.4–28.3%), and syphilis (4–5%) [5,6,8,9,10]. Voluntary nonremunerated blood donors (VNRBD) have also been identified as an important source for reliable and safe blood, and in 2018, the average proportion for this type of donor was 71% for African countries [5]. The inclusion of highly sensitive assays for TTIs has further increased the quality of blood. In the mid to late 1990s [11], additional testing that uses virus nucleic acid amplification technology to screen blood donors was introduced in most developed countries, thus, decreasing the residual risk of these infections over the years [11]. In addition to these molecular tests, approximately two decades ago, pathogen reduction technology (PRT) was introduced in blood banks in a few developed countries, reducing the residual risk of TTI to almost zero by inactivating pathogens. Inactivation is performed through several procedures that treat whole blood, plasma, or platelets with specific products such as detergents, solvents, ultraviolet light, and riboflavin [12,13]. In most SSA countries, TTI screening in blood banks is still conducted using rapid diagnostic tests (RDTs) [3,14,15]. Often, owing to the low sensitivity of these RDTs, they can miss infections during pre-conversion “window periods” and occasional occult hepatitis B (OBI) infections, characterised by the presence of HBV DNA in the absence of hepatitis B surface antigen (HBsAg), the principal serological marker for HBV infection [16]. The National Blood Institute in Angola, as in most SSA countries, uses RDTs to screen donors in their blood banks. Even though this country has recently introduced more sensitive tests (automated enzyme immunoassays), the RDTs are still used exclusively for donor screening in many regions distant from large centres, as reported by the National Blood Institute of Angola [17] (personal communication, 30 August 2021). The main objective of this study was to investigate the prevalence of HBV, HCV, HIV, and syphilis in blood donors at the *Hospital Geral do Bié*, Angola from 2005 to 2020. The information obtained may help assess screening programs and efforts to control TTIs and identify chronic HCV carriers and possible cases of OBI among blood donors in Angola. The results may additionally subsidise cost-benefit evaluations for the introduction of PRTs in blood banks in low-income countries.

## 2. Results

### 2.1. Blood Donor Profile

Between 2005 and 2020, 57,979 blood donors were assessed, of which most donations occurred between 2016 and 2020 (70.7%). Demographic data on sex and age were missing in 2.6% (1523/57,979) and 2.5% (1459/57,979) of blood donor records (Table 1). The year 2005 accounted for most of the missing information on sex (75.4%; 1100/1459) and age (68.4%; 1041/1523). The average proportion of blood donors screened for TTIs was 98.2% for HBV and HIV, 97.3% for syphilis, and 95.3% for HCV. Men represented the majority (*n* = 41,414; 73.4%), as well as the age range of 25–49 years (*n* = 30,040; 53.1%). Among the 57,979 registered blood donors, 10,172 TTIs were identified with an overall prevalence for HBV, HCV, HIV, and syphilis being 8.4% (95% CI: 8.2–8.7), 3.0% (95% CI: 2.8–3.1), 2.1% (95% CI: 2.0–2.2), and 4.4% (95% CI: 4.2–4.6), respectively (Table 2).

### 2.2. Trends over Time and Demographic Characteristics

The overall prevalence of TTIs fluctuated during the study period (Figure 1). During the 2014 voluntary non-remunerated blood donor campaigns, the prevalence of HBV, HIV, and syphilis, but not for HCV, decreased, with significant reductions in HIV (*X*^2^ = 23.58, *n* = 7147, *p* < 0.00001) and syphilis (*X*^2^ = 29.47, *n* = 6931, *p* < 0.00001). In 2017, significant reductions were observed in HBV (*X*^2^ = 134.22, *n* = 15,715, *p* < 0.00001), HCV (*X*^2^ = 10.87, *n* = 15,713, *p* = 0.001), HIV (*X*^2^ = 37.67, *n* = 15,715, *p* < 0.00001), and syphilis (*X*^2^ = 34.57, *n* = 15,715, *p* < 0.00001) (Figure 2).

The HBV prevalence was significantly higher among men (9.0%) than among women (6.8%) (*X*^2^ = 72.32, *n* = 55,568, *p* < 0.0001) (Table 3). Women donors had a significantly higher overall prevalence of HCV, HIV, and syphilis (*X*^2^ = 31.68, *n* = 54,045, *p* < 0.00001; *X*^2^ = 6.66, *n* = 55,442, *p* = 0.009; and *X*^2^ = 83.16, *n* = 54,048, *p* < 0.00001, respectively) (Table 3). Figure 3 shows the gender predominance of TTIs during the study period. The prevalence of all four TTIs gradually increased with age, with donors aged 50 years and older being significantly higher than those in age groups 18–24 years (*p* < 0.0001) and 25–49 (*p* < 0.0001) (Table 3 and Figure 4).

### 2.3. Blood Donor Specimens

#### 2.3.1. HCV

Among the 1500 blood donors registered in 2007, 500 serum samples were obtained for further testing, 412 were men and 88 women, with ages ranging from 18 to 64 years (average 29.3 ± 9.2). Positivity for anti-HCV was identified in 20 (4%) blood donors who had undetectable HCV RNA, and 4/20 (20%) were positive in the second anti-HCV assay. Ninety percent were men donors, and their ages ranged from 18 to 39 years. Malaria was detected in 14/20 (70%) of the anti-HCV positive samples (Appendix A).

#### 2.3.2. HBV-OBI

From the 500 serum samples from 2007, 170 residual samples were identified, with known negative results for TTIs, and volumes greater than 2 mL. HBV DNA was detected in 10/170 (5.8%) samples, with viral loads ranging from <10–37,652,193 IU/mL. Viral loads of <10 IU/mL were retested to exclude possible carry-over contamination, yielding identical results. The quantitative serological assay detected HBsAg in 5/10 samples, with concentrations varying from 0.05 to >150 IU/mL, three of which were likewise positive in the second HBsAg RDT (VIKIA^®^HBsAg/bioMérieux, Marcy-l’Étoile, France) assay. Anti-HBc was detected in 4/5 samples of HBsAg+/HBV DNA+. The remaining five (2.9%) samples were classified as possible OBI, as they were HBV DNA+/HBsAg-, with anti-HBc IgG positive in two samples (Table 4). Anti-HBc IgM testing was performed in all three samples, all of which were negative.

## 3. Discussion

### 3.1. Blood Donors

The increase in blood donations from 2016 to 2020 was mainly due to the high demand for blood transfusions to treat *falciparum* malaria anaemia, the leading cause of death in Angola. This disease, accounting for blood transfusion needs, rose sharply in 2016 and increased gradually thereafter. From 2005 to 2008 and 2013 to 2015, a total of 9,643,228 malaria cases, with 21,595 deaths, were registered compared with 29,591,660 cases and 61,368 deaths registered between 2016 and 2020. A three-fold increase in malaria cases and deaths and over a two-fold increase in blood donations (16,985 versus 40,984) were observed for the period 2016–2020. In addition to the rise in cases of malaria in 2016, Angola experienced a yellow fever epidemic during the same period, contributing to the high toll of deaths, overwhelming the health system, and consequently the need for large volumes of blood supplies, according to J. Martins, coordinator of the National Malaria Program [18]. In 2020, despite the high number of malaria cases, a decrease in the number of transfusions was registered. This decrease was due to the change in treatment protocol in 2019, substituting quinine for artesunate as the first choice to treat severe cases of malaria, as the latter quickly clears circulating parasites. Thus, patients are less likely to have haematological complications that require blood transfusion [19,20].

During the period 1992–2002, the civil war in Angola devastated the country, especially in the central and southern regions of the province of Bié, where this study took place [21]. In 2005, when the study was initiated, fragility in data collection was noted in the blood bank records, with missing demographic data detected in 76.92% of the data. Following that year, data registration improved with the following percentage of missing data: 2006 = 1.84%, 2007 = 6.20%, 2008 = 6.03%, 2013 = 0.00%, and 2014 = 3.64%. In the second semester of 2014, the National Blood Institute of Angola defined and implemented methodological guidelines, which improved data recording, especially after 2015, with no missing demographic data observed after this period (Decree Nº 281/14). Regarding missing laboratory results, before 2015, blood donor screening for HBV, HCV, HIV, and syphilis was not 100%, despite the WHO recommendation that all blood donors be tested for infectious diseases. One might consider missing testing data due to failure to register the test results. However, according to laboratory personnel, this was less likely to have occurred as the Hospital Geral do Bié´s blood bank implemented a local testing algorithm to overcome diagnostic test shortages. HBV was the initial test for screening blood donors; those testing negative were further screened sequentially for HIV, HCV, VDRL, and malaria. If any test was reactive, further screening ceased.

The vast majority of donors were men (73.4%), as observed by other authors in different sub-Saharan African countries, such as in the Democratic Republic of Congo (70.3%) [22], Burkina Faso (75.6%) [23], Equatorial Guinea (76.8%) [24], Cameroon (78.3%) [25], Malawi (80%) [26], Sierra Leone (80%) [27], Gabon (83%) [14], Tanzania (83.7%) [28], Mali (88.8%) [29], Ghana (92.2%), Mauritania (95%), and Nigeria (98.7%) [30,31,32]. This male donor predominance is commonly reported in low- and middle-income countries [33] and is attributed to their availability for supplying immediate blood to their family, friends, or neighbours that require a blood transfusion.

### 3.2. HBV

Sub-Saharan Africa is endemic for HBV and has an estimated overall prevalence of over 8% in the general population [34]. The prevalence of hepatitis B in blood donors depends on the number of infected individuals with hepatitis B in the general population and the type of donor. Replacement donors are said to have a higher prevalence when compared to VNRBD; the latter is identified as providing the safest blood, as most are at low risk for infectious diseases [33]. Since replacement donors are the majority in Angola, the results could reflect the overall country seroprevalence. The prevalence of HBV in blood donors in Angola reported by Apata et al. [35] for the period 2000/2004 (8.68%) was similar to the average prevalence detected for 2005/2008 and 2013/2020 in this study (8.5%). Apata et al. reported a decrease in HBV prevalence in 2010/2011 [35], a period in which blood donor records were unavailable for this study. Nonetheless, a reduction in HBV prevalence was observed in this study in 2014 and 2017, when the Provincial Health Office of Bié conducted numerous campaigns for VNRBD. In addition to Angola, high seroprevalences for HBV have been similarly observed in blood donors in at least 11 SSA countries: 9.2% in the Democratic Republic of Congo, 9.7% in Sierra Leone, 10% in Equatorial Guinea, 10.1% in Chad, 10.5% in Senegal, 10.6% in Mozambique, 11.8% in Niger and Mauritania, 14.8% in Mali, 15% in Burkina Faso, and 46.8% in Benin [23,29,36,37].

### 3.3. HCV

HCV prevalence is estimated to be above 3.5% in North Africa and 1.5–3.5% in Sub-Saharan Africa [34,36]. In this study, HCV was 3%, which was within the estimated standards for the region. Low prevalence was found in other sub-Saharan African countries: 1% in Malawi [26], 0.9% in Namibia [38], and 0.3% in Eritrea [39], and the highest prevalence was reported in Burkina Faso and Ghana [23,30], ranging from 7 to 9.4%.

In this study, a gradual increase in the prevalence of HCV was noted over the years, which may reflect not only the use of improved sensitivities in the RDT, but the fact that the province of Bié was isolated from the country and the world from 1992 to 2002; however, with definitive peace, there were demining and improvements in inland routes, which made access to this province easier. Following these improvements, the flow of individuals from within the country and neighbouring countries intensified, which may have likewise contributed to the increased prevalence of anti-HCV in the period analysed.

### 3.4. HIV

HIV infection remains a global public health concern, especially in Africa, particularly in the western and central regions, where 21% of new HIV infections are reported globally [29]. Blood transfusion is responsible for an estimated 1% of new HIV infections [40], and screening of blood donors in this region is predominantly performed with RDT. The performance of the latest RDTs for HIV is equivalent to automated enzyme-linked immunosorbent assays, as shown in several studies evaluating rapid HIV tests for transfusion screening in Africa, which reported sensitivities of approximately 100%, supporting its use for screening blood donors [3]. In Angola, the HIV prevalence in the general population is 2% [2,41], and the blood donor prevalence during the study period was identical (2.1%). Similar results were reported in Malawi; (1.9%) [26]. Higher HIV seroprevalence has been reported in Mozambique (8.5%) and Equatorial Guinea (7.83%) [14] and is related to the high prevalence in the general population. The Hospital Geral do Bié´s blood bank uses RDT for blood donor screening, but since December 2019, additional testing for HBV, HCV, and HIV has been performed using an automated immunofluorescence assay (MINI VIDAS^®^/bioMérieux S.A., Marcy-l’Étoile, France) to lower residual risk. The present national screening algorithm requires that all blood candidates be screened first by a RDT, and those with negative results will be tested using the commercial automated MINI VIDAS^®^/bioMérieux assay. The prevalence of HIV in 2020 was comparable to the general prevalence observed over the study period, which supports the high specificity and sensitivity attributed to RDT for HIV and the improved performance in the testing assays over the years [3].

### 3.5. Syphilis

Syphilis remains a severe public health problem globally, with an incidence of 0.5 cases per 1000 inhabitants and a high prevalence in the African region, which requires extended precautions related to blood safety [14,42]. In this study, the prevalence of syphilis in blood donors was 4.4%, similar to that reported in Burkina Faso (3.9%) [23]. Equatorial Guinea (21.5%) and Tanzania (12.7%) were registered with the highest prevalence in the SSA region [14], and the lowest prevalence was found in Ethiopia (1.1%) and Mali (0.01%) [29,40]. These variations in prevalence may be due to differences in sensitivity and specificity of screening tests, preventive measures, the effectiveness of the screening program, choice of donors, and accessibility to healthcare [14,42,43].

The varied prevalence observed in this study for the TTIs was either higher or within the WHO estimates for low-income countries [33]. While HIV values were mainly within the interquartile range (Figure 5), HCV and syphilis prevalence were typically above these ranges. In contrast, HBV prevalence was 50% higher and 50% within the WHO range.

The seroprevalence variations observed along the years in this study for HBV, HCV, HIV and syphilis infections might be attributed to the different analytical sensitivities in the tests used during the study period (Appendix A).

### 3.6. TTIs and Demographic Characteristics

The prevalence of HBV was higher in men, similar to that observed in other countries in the region, such as Gabon, Ethiopia, Mali, and South Africa [14,29,40,44]. This finding can be explained by the fact that men are more prone to progress to chronic HBV infection, with an expected difference in sex distribution for chronic HBV carriers [45,46]. In contrast, women blood donors had a higher prevalence of HCV than that found in Gabon, where prevalence was similar in both sexes [14]. In Ethiopia, however, the prevalence of HCV infection is higher in men [40]. Women had a higher prevalence of HIV blood donors than men as reported in Angola [2,41], and this may be partially explained by the characteristics of the genital mucosa of women, which is subject to a greater risk of contracting the infection [28]. Globally, the number of new cases of HIV registered in 2019–2020 was similar for both sexes [1,2]; however, when limited to the SSA region, women had a higher number of new HIV infections than men [1]. Similar to HCV and HIV, syphilis had a higher prevalence in women donors (5.8 versus 3.9%) as reported in Namibia [38], although most SSA countries have demonstrated a higher prevalence of syphilis among men donors; Ethiopia, Cameroon, Gabon, Tanzania, Eritrea, Kenya, and Nigeria [14,25,28,32,40,47,48].

The seroprevalence of TTIs is associated with age, as there is more exposure to infections over time, which consequently results in a higher prevalence in advanced ages [40]. Studies conducted in Ethiopia support this association, and blood donors over 50 years old, in this study, had the highest prevalence for all four infections: HBV, HCV, HIV, and syphilis [40]. Studies in Mauritania showed the highest prevalence in this age group for HBV, HCV, and HIV; Burkino Faso, Eritrea, and Gabon for HIV and syphilis; Mali for HIV and HCV [14,23,29,31,47]; and Tanzania for HCV and syphilis [28]. Furthermore, South Africa had the highest prevalence of HCV in blood donors over 60 years old [44], suggesting possible parenteral exposure in the past.

Besides the fact that age is associated with infection exposure, there are additional reasons to justify the elevated prevalence of syphilis with age. The non-treponemal tests used frequently in blood banks in the African region may cross-react with antibodies from other acute febrile and autoimmune diseases, resulting in false-positive results. Another factor is the treponemal antibody test that remains positive for life in 85% of patients after treatment. A positive treponemal antibody test does not differentiate active infection from the past [42]; therefore, this persistence of antibodies in the absence of infection elucidates the increase in syphilis prevalence with increasing age of blood donors [14].

### 3.7. Blood Donor Specimens

#### 3.7.1. HCV: Blood Donor Specimens

Of the 500 blood donor specimens collected in 2007, 4% tested positive for anti-HCV and lacked detectable virus, confirming a low prevalence of chronic HCV and translating as past infection or cross-reactivity. Spontaneous clearance of HCV occurs in at least 25% of infected individuals, and anti-HCV antibodies may be detectable in the absence of the virus [49]. Although most of the anti-HCV-positive samples from this group tested positive for malaria, speculations have been made regarding false-positive anti-HCV tests and other infectious diseases [50], but no study has confirmed this association. Most seroprevalence studies in blood banks use anti-HCV as the sole marker for HCV prevalence in SSA; however, Vermeulen and collaborator-based South African blood donors have reported molecular detection of HCV RNA, with an overall prevalence of 0.03, comparable to the WHO estimates [33,44].

#### 3.7.2. HBV: Occult B Infection

The blood bank in Bié, Angola, had used the same screening policy for HBV for over 15 years, with RDT as their main screening assay for HBsAg until 2019, when the commercial automated immunoassay system MINIVIDAS®/bioMérieux was introduced.

In this study, HBV DNA testing was able to detect blood donors with HBV infection missed by the standard screening protocol of Angola. The highly sensitive quantitative commercial HBsAg assay was unable to detect HBV infection in 3% (5/170), identified as possible OBI, and was within the range of <1–87%, reported in other geographic regions, but lower than what has been reported in other SSA countries that are highly endemic for hepatitis B (Nigeria, 17%; Sudan, 38%) [51,52]. As second-time samples were not available, one cannot rule out the prospect of samples being collected during an acute phase window period, when HBsAg is not detected before its appearance, especially for the three samples that were negative for anti-HBc IgG and IgM. Therefore, residual risk remains an issue for HBV and other TTIs, despite the introduction of sensitive commercial assays. High-income countries that have implemented NAT to screen blood donors report residual risks for TTIs to be <1 per 1 million units of blood tested [53,54,55]. In SSA countries, where NAT testing is scarce, a different scenario is observed with unsettling high residual risks for these infections reported that vary from 67 to 250 per 1 million units tested [56]. Concerns remain with emerging pathogens not routinely tested in blood banks that jeopardise the quality of blood [13]. Pathogen inactivation through different PRT in plasma and platelet are already widespread in high-income countries. However, whole blood PRTs are not yet commercially available but are presently undergoing clinical trials [55,56]. Considering the high demand for whole blood and elevated prevalences of TTIs in SSA countries, the effective use of PRT would be an opportunity to increase safe blood availability. A PRT for whole blood could revolutionise blood safety in countries that lack NAT testing and additionally prevent the widespread of new emerging infections through blood transfusions.

## 4. Materials and Methods

### 4.1. Study Populations

#### 4.1.1. Blood Donor Data

A retrospective cross-sectional study was conducted at the Hospital Geral do Bié, Cuito, Bié Province, Angola, to analyse blood donor data from January 2005 to December 2008, and from January 2013 to December 2020. Blood donor records between 2009 and 2012 were unavailable; therefore, data from this period were absent. Variables, including date of donation, age, gender, and screening test results for HBV, HCV, HIV, and syphilis, were retrieved from paper documents. Age groups were categorised according to the criteria of the National Institute for Combating AIDS, Angola into the following age groups: 18–24, 25–49, and >50, according to O. Machado (personal communication, 30 August 2021), director of the National Blood Institute. This study ranked two periods: before 2016 (years 2005, 2006, 2007, 2008, 2013, 2014, and 2015) and after year 2016 (years 2016, 2017, 2018, 2019, and 2020). The year 2016 was selected because there was a sharp increase in blood collection following outbreaks of several infectious diseases with high demands for blood and blood products [57]. The prevalence of each TTI was obtained by dividing the number of reactive donors by the total number of blood donors tested for each year for each specific infection. When estimating prevalence according to sex or age category, missing data were excluded for calculation purposes.

#### 4.1.2. Blood Specimens

Frozen serum samples from 500 consecutive previously consenting blood donors at the HGB between February and July 2007 were investigated for HCV using serological and molecular tests. Following these tests, residual samples with volumes greater than 2 mL were selected to explore possible OBI. As the study used one-time blood donor-stored samples, OBI was identified only as possible and not confirmed cases. This is because HBsAg may be similarly absent during the pre-conversion window period of acute HBV infection. Samples collected in 2007 were centrifuged, serum aliquoted into two cryotubes, and stored in a freezer at −30 °C at the HGB until they were sent to Brazil. The samples were shipped on dry ice to the Viral Hepatitis Laboratory/Oswaldo Cruz Institute/Oswaldo Cruz Foundation/Brazil and immediately stored at –80 °C until thawed for serological or molecular tests for HCV and HBV. 

### 4.2. Serological Marker Detection Using Rapid Diagnostic Tests (RDT) at the HGB

Blood donors were tested following the testing policy at the HGB that established HBV for initial screening; those testing negative were further screened sequentially for HIV, HCV, syphilis, and malaria. The testing was performed using different RDTs according to availability. HBsAg RDT was used for HBV detection, with HBsAg Determine^®^ (Abbott Japan Co., Ltd., Tokyo, Japan) as the principal commercial kit used throughout the study period. RDTs for antibody detection were used for HCV, HIV, and syphilis. Anti-HCV Spot^®^ (Genelabs Diagnostics Pty Ltd, Singapore) is a major screening RDT for HCV testing, and HIV 1/2 Determine^®^ (Abbott Japan Co., Ltda., Tokyo, Japan) and Uni-Gold™ HIV (Trinity Biotech Plc., Bray, Ireland) were the main tests used for HIV screening in blood donors. Syphilis test results, besides RDT, were also based on non-treponemal antibody tests, the rapid plasma reagin (RPR) assay, and the Venereal Disease Research Laboratory (VDRL) assay. A range of diagnostic tests was used for each of the TTIs and is listed in Table Supplement 2, according to availability. 

### 4.3. Serological Markers and Molecular Testing for Hepatitis in Brazil

#### 4.3.1. HCV Testing

The 500 blood donor specimens were tested using a commercial enzyme-linked immunosorbent assay, HCV Ultra Hepanostika^®^ (bioMerieux S.A., Craponne, France), according to the manufacturer´s instructions, and were repeated in duplicate and considered positive if all results were reactive. Reactive specimens for anti-HCV were investigated for HCV RNA using a commercial quantitative PCR assay (m2000rt Abbott RealTime HCV^®^ Abbott Molecular Inc., Des Plaines, IL, USA), and samples with undetectable HCV RNA were additionally tested for anti-HCV using a second assay (CHIRON RIBA HCV 3.0 Strip Immunoblot Assay/Chiron Corporation, Emeryville, CA, USA).

#### 4.3.2. Malaria Testing

To identify possible false-positive anti-HCV results, additional screening for *Plasmodium falciparum* and *Plasmodium vivax* malaria were included and performed using an RDT, SD BIOLINE^®^Malaria Ag Pf/Pan Standard Diagnostics, Inc., Seoul, Korea.

#### 4.3.3. HBV Testing

To explore possible OBI, residual samples from 500 blood donors collected in 2007, with volumes greater than 2 mL and known negative results for HBV, HIV, HCV, and syphilis were selected to investigate the presence of HBV DNA using a commercial quantitative PCR assay (Abbott RealTime HBV^®^ Abbott Molecular Inc., Des Plaines, IL, USA). Samples with detectable HBV DNA were retested for HBsAg using a different commercial RDT (VIKIA^®^HBsAg/bioMérieux, Rio de Janeiro, Brazil), and a quantitative HBsAg assay (Liaison^®^ XL HBsAg Quant/DiaSorin, Saluggia, Italy), in addition to the anti-HBc testing (IgG and IgM) performed with commercial assays (Liaison XL^®^/DiaSorin S.p.A., Italy).

### 4.4. Statistical Analysis

We estimated prevalence as the proportion of individuals who had a positive result in the HBsAg, anti-HCV, anti-HIV, or syphilis tests. The data were stratified using the year of occurrence, sex, age, and disease. The 95% confidence intervals were calculated using the Clopper–Pearson method. Analyses were performed using the R Studio statistical software (version 1.4). Statistical significance corresponds to P-values less than 0.05.

## 5. Conclusions

During the study, we found significant improvements in (1) blood bank input recording; (2) screening coverage for HBV, HCV, HIV, and syphilis; and (3) updated screening algorithms that included assays with higher analytical sensitivities. The decrease in TTI prevalence noted during VNRBD campaigns reinforces the importance of this type of donor to ensure safe blood. HBV has the highest burden of TTI in Angola and needs to be addressed with care, as it is a vaccine-preventable disease. Adult vaccination for HBV and treatment needs to be implemented by Angola´s public health system to reduce new infections, which may be missed in the window periods by the present blood donor screening algorithm. Considering the high demand for whole blood and elevated prevalences of TTIs and emerging pathogens in SSA countries, the effective use of PRT in whole blood, once commercially available, will be an opportunity to increase safe blood availability.

## Figures and Tables

**Figure 1 pathogens-10-01633-f001:**
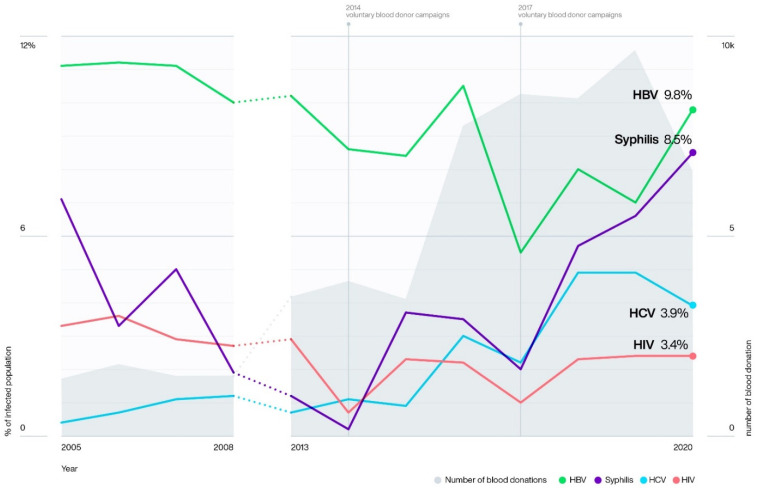
Variations in seroprevalence of HBV, HCV, HIV, and syphilis and number of blood donations registered during the periods: 2005 to 2008 and 2013 to 2020 in Bié, Angola. Dotted lines (…..) correspond to the time period 2009 to 2012, when no blood donor data was available.

**Figure 2 pathogens-10-01633-f002:**
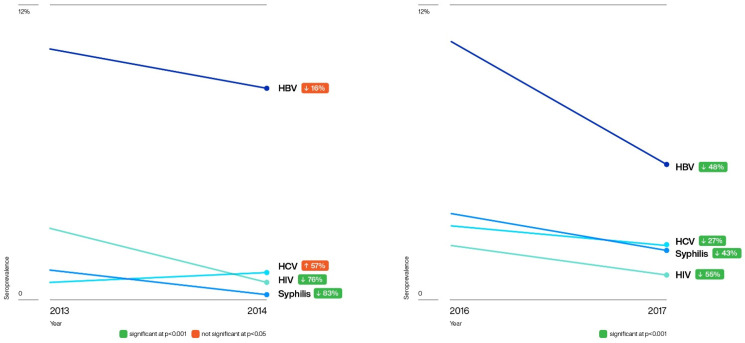
The seroprevalences of HBV, HCV, HIV, and syphilis were compared before (2013 and 2016) and with intensified blood donation campaigns (2014 and 2017), using the Chi-square test. Significant differences (*p* < 0.001), green colour, were observed for HIV and syphilis between 2013 and 2014 and 2016 and 2017 for all infections. The red colour denotes not significant.

**Figure 3 pathogens-10-01633-f003:**
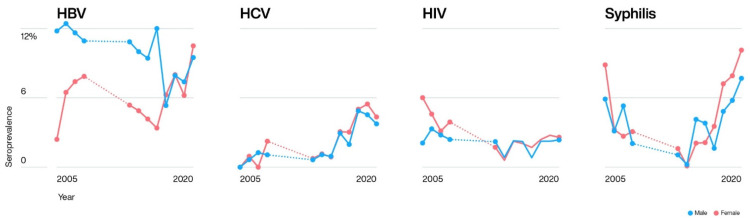
Gender distribution according to the transfusion transmissible infections. The blood donor seroprevalences for males are marked in blue and females with red. Dotted lines (…..) correspond to the time period 2009 to 2012, when no blood donor data was available.

**Figure 4 pathogens-10-01633-f004:**
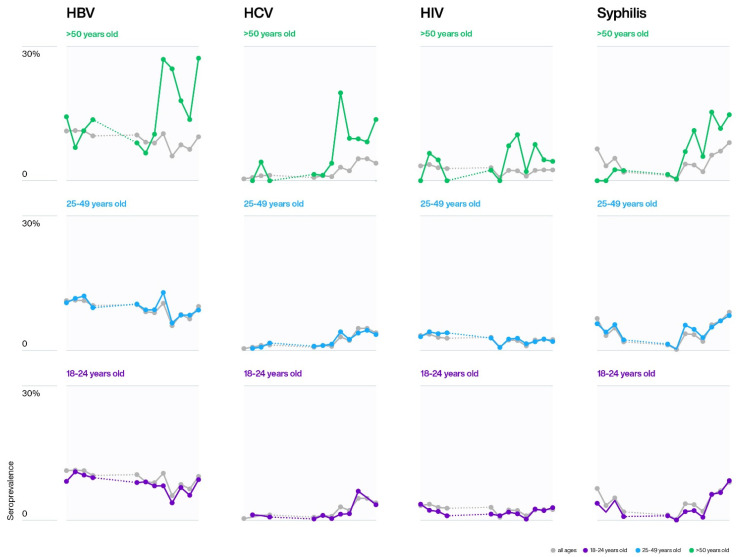
The TTI seroprevalences for the age groups: 18–24 (purple), 25–49 (blue), >50 (green) were compared with the annual average (grey). Dotted lines (…..) correspond to the time period 2009 to 2012, when no blood donor data was available. The Chi-square test showed significant (*p* <0.0001) higher prevalences in the age group >50 when compared to the other two age groups.

**Figure 5 pathogens-10-01633-f005:**
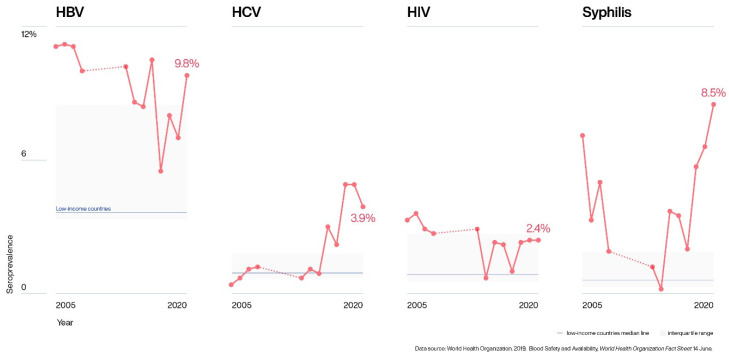
Representation of the variation of transfusion-transmitted infections over the study period, compared with the seroprevalence median of blood donors established by the WHO for low-income countries [33]. Dotted lines (…..) correspond to the time period 2009 to 2012, when no blood donor data was available.

**Table 1 pathogens-10-01633-t001:** General data on donors and testing in Bié.

Year	Blood Donors ^a^ (%)	Age Groups ^b^				Gender ^c^			Number of Blood Donors Tested ^d^			
		18–24 (%)	25–49 (%)	>50 (%)	Unknown	Male (%)	Female (%)	Unknown	HBsAg (%)	Anti-HCV (%)	Anti-HIV (%)	Anti-Treponema (%)
2005	1430 (2.5)	115 (34.8)	208 (63.0)	7 (2.1)	1100	306 (78.7)	83 (21.3)	1041	1429 (99.9)	1135 (79.4)	1371 (95.8)	1232 (86.2)
2006	1792 (3.1)	647 (36.8)	1058 (60.1)	54 (3.1)	33	1417 (79.8)	359 (20.2)	16	1780 (99.3)	1554 (86.7)	1737 (96.9)	1630 (91)
2007	1500 (2.6)	616 (44)	747 (53)	44 (3.1)	93	1254 (88.6)	162 (11.4)	84	1490 (99.3)	848 (56.5)	1463 (97.5)	1372 (91.5)
2008	1508 (2.6)	601 (42.4)	772 (54.5)	44 (3.1)	91	1138 (89.7)	130 (10.3)	240	1498 (99.3)	1054 (69.9)	1487 (98.6)	1486 (98.5)
2013	3476 (6)	1670 (48)	1581 (45.5)	225 (6.5)	0	2490 (71.6)	986 (28.4)	0	3307 (95.13)	3279 (94.33)	3278 (94.30)	3279 (94.33)
2014	3869 (6.6)	1622 (43.5)	1847 (49.5)	259 (7)	141	2726 (73.1)	1002 (26.9)	141	3750 (96.9)	3679 (95)	3869 (100)	3652 (94.4)
2015	3420 (5.9)	1790 (52.3)	1551 (45.3)	79 (2.3)	0	2730 (79.8)	690 (20.2)	0	3331 (97.4)	3331 (97.4)	3330 (97.36)	3331(97.4)
2016	7751 (13.4)	3926 (50.6)	3709 (47.8)	116 (1.5)	0	6379 (82.3)	1372 (17.7)	0	7171 (92.5)	7169 (92.4)	7171 (92.5)	7171 (92.5)
2017	8544 (14.7)	3585 (41.9)	4811 (56.3)	148 (1.7)	0	6964 (81.5)	1580 (18.5)	0	8544 (100)	8544 (100)	8544 (100)	8544 (100)
2018	8436 (14.6)	2904 (34.4)	5297 (62.8)	235 (2.8)	0	5388 (63.9)	3048 (36.1)	0	8436 (100)	8436 (100)	8436 (100)	8436 (100)
2019	9640 (16.6)	4568 (47.4)	4772 (49.5)	300 (3.1)	0	6168 (64)	3472 (36)	0	9640 (100)	9640 (100)	9640 (100)	9640 (100)
2020	6613 (11.4)	2641 (39.9)	3687 (55.8)	285 (4.3)	0	4454 (67.4)	2159 (32.6)	0	6613 (100)	6613 (100)	6613 (100)	6613 (100)
Total	57,979 (100)	24,685 (43.7)	30,040 (53.1)	1796 (3.2)	1459	41,414 (73.4)	15,043 (26.6)	1523	56,961 (98.2)	55,282 (95.3)	56,939 (98.2)	56,386 (97.2)

^a^ total number of blood donors registered for each year, with percentages related to the distribution for the study period; ^b^ age groups are expressed in years and frequency calculated only for known ages; ^c^ frequencies calculated exclusively for known gender; ^d^ number of blood donors tested for each TTI with percentages calculated using the total number 6of blood donors registered.

**Table 2 pathogens-10-01633-t002:** TTI seroprevalence for the years 2005/2008 and 2013/2020.

	HBV+				HCV+				HIV+				Syphilis+			
Year	Total	*n*	%	CI	Total	*n*	%	CI	Total	*n*	%	CI	Total	*n*	%	CI
2005	1429	159	11.1	9.5–12.8	1135	4	0.4	0.1–0.8	1370	45	3.3	2.4–4.3	1232	88	7.1	5.7–8.7
2006	1780	200	11.2	9.8–12.7	1554	11	0.7	0.3–1.2	1712	61	3.6	2.7–4.5	1630	54	3.3	2.4–4.3
2007	1490	166	11.1	9.5–12.8	848	9	1.1	0.4–2.0	1458	42	2.9	2.0–3.8	1372	69	5	3.9–6.3
2008	1498	150	10	8.5–11.6	1054	13	1.2	0.6–2.0	1489	40	2.7	1.9–3.6	1486	28	1.9	1.2–2.7
2013	3279	305	9.3	8.3–10.4	3279	22	0.7	0.4–1.0	3278	68	2.1	1.6–2.6	3279	40	1.2	0.8–1.6
2014	3750	322	8.6	7.7–9.5	3679	39	1.1	0.7–1.4	3869	29	0.7	0.5–1.0	3652	7	0.2	0.1–0.3
2015	3331	279	8.4	7.4–9.3	3331	31	0.9	0.6–1.3	3330	76	2.3	1.8–2.8	3331	124	3.7	3.1–4.4
2016	7171	752	10.5	9.7–11.2	7169	215	3	2.6–3.4	7171	157	2.2	1.8–2.5	7171	252	3.5	3.0–3.9
2017	8544	471	5.5	5.0–6.0	8544	185	2.2	1.8–2.4	8544	84	1.0	0.8–1.2	8544	170	2	1.7–2.3
2018	8436	672	8	7.5–8.7	8436	416	4.9	4.4–5.4	8436	194	2.3	2.0–2.7	8436	480	5.7	5.3–6.3
2019	9640	672	7	6.4–7.4	9640	470	4.9	4.4–5.3	9640	235	2.4	2.1–2.7	9640	632	6.6	6.0–7.0
2020	6613	650	9.8	9.1–10.5	6613	261	3.9	3.4–4.4	6613	161	2.4	2.0–2.8	6613	562	8.5	7.8–9.1
Total	56,961	4798	8.4	8.2–8.7	55,282	1676	3	2.8–3.1	56,910	1219	2.1	2.0–2.2	56,386	2506	4.4	4.2–4.6

TTI, transfusion transmissible infection; Total, refers to the number of blood donors tested for each infection; *n*, number of blood donors that tested positive; %, percentage; HBV, hepatitis B virus; HCV, antibodies to hepatitis C virus; HIV, human immunodeficiency virus; syphilis, antibodies to *Treponema palladium*; CI, 95% confidence interval.

**Table 3 pathogens-10-01633-t003:** Frequency of positive cases by gender and age for the study period. Statistical significance at *p* < 0.0001 for most data, unless indicated.

Demographics	Number	HBV				HCV				HIV				Syphilis			
		Total	*n*	%	OR 95% CI	Total	*n*	%	OR 95% CI	Total	*n*	%	OR 95% CI	Total	*n*	%	OR 95% CI
Age																	
18–24	24,684	24,242	1734	7.2	1	23,631	657	2.8	1	24,194	418	1.7	1	24,034	886	3.7	1
25–49	30,040	29,615	2626	8.9	1.26 (1.18–1.34)	28,796	881	3.1	1.10 (0.99–1.22) ^a^	29,547	656	2.2	1.29 (1.14–1.46)	29,331	1391	4.7	1.30 (1.19–1.42)
>50	1796	1773	285	16.1	2.48 (2.17–2.85)	1725	132	7.7	2.89 (2.39–3.52)	1765	76	4.3	2.56 (1.99–3.28)	1746	144	8.3	2.35 (1.96–2.82)
Gender																	
Male	41,414	40,709	3648	9.0	1.35 (1.26–1.46)	39,399	1114	2.8	0.73 (0.67–0.82)	40,592	801	2.0	0.84 (0.74–0.96)	40,276	1580	3.9	0.66 (0.61–0.73)
Female	15,043	14,859	1003	6.8	1	14,646	555	3.8	1	14,850	346	2.3	1	14,772	853	5.8	1
Gender/Age																	
Male																	
18–24	18,376	18,011	1369	7.6	1	17,492	457	2.6	1	17,967	299	1.7	1	17,841	547	3.1	1
25–49	21,709	21,385	2040	9.5	1.28 (1.19–1.38)	20,715	567	2.7	1.04 (0.93–1.19) ^a^	21,322	450	2.1	1.27 (1.10–1.48)	21,150	932	4.4	1.46 (1.31–1.62)
>50	1124	1111	211	19.0	2.85 (2.43–3.34)	1069	88	8.2	3.34 (2.64–4.21)	1103	45	4.1	2.51 (1.83–3.46)	1089	90	8.3	2.84 (2.26–3.59)
Female																	
18–24	6179	6102	358	5.9	1	6029	200	3.3	1	6099	118	1.9	1	6066	338	5.6	1
25–49	8170	8073	571	7.1	1.22 (1.07–1.40)	7950	311	3.9	1.18 (0.99–1.42) ^a^	8067	196	2.4	1.26 (1.0–1.59)	8028	456	5.7	1.02 (0.88–1.18) ^a^
>50	662	652	73	11.2	2.02 (1.55–2.64)	646	44	6.8	2.13 (1.52–2.98)	652	31	4.8	2.53 (1.69–3.79)	647	54	8.4	1.54 (1.14–2.08)

Number, refers to the total number of blood donors registered from 2005 to 2008 and 2013 to 2020; HBV, hepatitis B virus; HCV, antibodies to hepatitis C virus; HIV, human immunodeficiency virus; syphilis, antibodies to *Treponema palladium*; Total, refers to the number of blood donors tested for each specific infection; *n*, number of blood donors that tested positive; %, percentage; OR, odds ratio; CI, 95% confidence interval; ^a^, no statistical significance.

**Table 4 pathogens-10-01633-t004:** Serological and molecular profiles in blood donor samples identified as possible OBI.

Number	ID	1st RDT	2nd RDT	qHBsAg	qHBsAg	Anti-HBc	Anti-HBc	HBV DNA
		Determine^®^	Vikia^®^	Liason^®^	IU/mL	IgG	IgM	IU/mL
1	241	*n*	*n*	*n*	0.036	*n*	*n*	272
2	249	*n*	*n*	*n*	<0.030	*n*	-	18
3	284	*n*	*n*	*n*	<0.030	*n*	*n*	<10
4	349	*n*	*n*	*n*	<0.030	*p*	-	13
5	389	*n*	*n*	*n*	<0.030	*p*	-	38

OBI, occult hepatitis B infection; ID, sample identification; RDT, rapid diagnostic test; qHBsAg, quantitative surface antigen test; anti-HBc, hepatitis B core antibody; IU/mL, international units per millilitre; IgG, immunoglobulin class G; IgM, immunoglobulin class M; *n*, negative; *p*, positive.

## Data Availability

Data reported in this study are available upon request. Datasets obtained from the Hospital Geral do Bié´s blood bank are not publicly disclosed as they contain confidential information, and thus in accordance with national regulations, must remain private.

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
