# Peer review of "Transfusion Transmissible Infections in Blood Donors in the Province of Bié, Angola, during a 15-Year Follow-Up, Imply the Need for Pathogen Reduction Technologies"

_pathogens, 2021, doi:10.3390/pathogens10121633_

Round 1

Reviewer 1 Report

Authors investigated transfusion transmissible infection such as HBV, HIV, HCV and syphilis in sub-Sahara Africa 2005-2020.

Although this manuscript is potentially interesting, several issues arise.

This manuscript is too long. Please summarize.

Please show the statistical difference TTIs between voluntary and non-voluntary donors.

Why was the prevalence of HBV high?

Why were different the prevalence of TTIs among each year?

Materials and Methods should be before Results.

Abstract In this time, this conclusion is not appropriate.

Table 1. This table is too busy.

        In No of blood donors What is (  )?

Table 2. Does HBV mean HBV positive?

        Authors should explain abbreviation in legend.

Figure 1 Y axis is not clear.

Figure 2 This figure is not clear and need explain in legend.

         Statistical analysis may be helpful.

Table 3 Authors should explain abbreviation in legend.

Figure 3 – 5. These figures are not clear and encouraged to be remade.  

              Authors should explain these figures in legend.   

Table 4 Authors should explain abbreviation in legend.

Author Response

Manuscript Title: Transfusion transmissible infections in the province of Bié, Angola: a 15-year follow-up

Corresponding Author: Lia Laura Lewis-Ximenez

Reviewer 1

Comments

# 1: Manuscript is too long

Answer: We have excluded a few lines as suggested by one of the reviewers (lines 183-190) in the discussion. We would, nevertheless, like to keep the remaining text in the manuscript as this large sample size retrospective blood bank study is unique in Angola. The results obtained will be able to subsidize government entities to implement public health decision-making policies towards improvements in blood safety. In addition to the results, information in the discussion section may offer guidelines for continued research in this study topic.

# 2: Please show the statistical difference TTIs between voluntary and non-voluntary data.

Answer: As the type of blood donor was not initially retrieved from the blood bank´s registration documents, we requested this data upon the reviewer´s inquiry and received raw data as a percentage for each year. We did observe higher percentages of voluntary donors in 2014 and 2017. However, as we do not have this data individually for each donor, it is impossible to carry out statistical analysis, thus a limitation in our study. We would also like to remind the reviewer that all data from the Hospital Geral do Bié´s blood bank is registered in paper documents.  

# 3: Why was the prevalence of HBV high?

Answer: This was explained in the discussion section, lines 192 to 199 when cited the high prevalence of HBV in sub-Saharan Africa´s general population. Furthermore, the high prevalence of HBV in blood donors in Angola and other countries reported by other authors was also mentioned. The authors also explain that the replacement donors, which are the majority in this region and in Angola, reflect the general populations´ prevalence of HBV.

# 4: Why were different the prevalence of TTIs among each year?

Answer:  A range of diagnostic tests for each TTI was used during the study period (supplement 2), each with different analytical sensitivities which may have contributed to these yearly variations. For the years 2014 and 2017, the authors describe the decrease in prevalence due to the intensified voluntary blood donor campaigns. The authors included in the Discussion Section a few sentences regarding diverse analytical sensitivities that could explain these variations. We have also included a sentence in the Discussion, under HCV, regarding test analytical sensitivity as we had initially explained the increase of anti-HCV positivity due to the increase of flow of individuals in the province after the end of the civil war.

# 5: Materials and Methods should be before Results.

Answer: According to the journal's recommendations and also in its publication template available in the "Instructions to Authors" area, the "Results" section must be presented before the "Materials and Methods" and "Discussion" sections. This information is described in the "Manuscript Preparation" section, more specifically in the "General Considerations" and "Research Manuscript Sections" subsections.

# 6:Abstract In this time, this conclusion is not appropriate

Answer: We did not understand the meaning of this comment, therefore unable to answer.

# 7: Table 1: This table is too busy.

                      In No of blood donors What is ( )?

Answer: Table 1 was restructured for better understanding and ( ) described in table.

# 8: Table 2. Does HBV mean HBV positive?

                     Authors should explain abbreviation in legend

Answer: Added the sign “+” after each TTI to represent positivity and included legend to explain abbreviations. Abbreviation for “number” originally No was corrected to N;  confidence interval had been erroneously abbreviated as IC, and was corrected to CI.

# 9. Figure 1 Y axis is not clear.

Answer: Y-axis title referring to blood transfusions was placed in the z-axis title for better understanding.

# 10. Figure 2. The figure is not clear and needs explain in legend. Statistical analysis may be helpful.

Answer: Figure 2 was better explained in the legend and statistical analysis included.

# 11. Table 3. Authors should explain abbreviation in legend.

Answer: Abbreviations included in the legend.

# 12. Figure 3 - 5. These figures are not clear and encouraged to be remade.

Authors should explain figures in legend.

Answer: Figures 3 to 5 were explained in more detail.

# 13: Table 4. Authors should explain abbreviation in legend.

Answer: Abbreviations included in the legend.

Reviewer 2 Report

The paper reports a retrospective cross-sectional study, conducted among blood donors in Angola, in which was evaluated the trend of transfusion transmissible infections (TTIs) caused by hepatitis B virus (HBV), human immunodeficiency virus (HIV), hepatitis C virus (HCV), and syphilis during a 15 year follow-up. In addition, frozen samples, obtained from blood donors in 2007, have been analyzed in order to identify chronic HCV carriers and possible occult HBV infection (OBI).  

Results show that:

  1. The overall prevalence of HBV, HCV, HIV and syphilis was 8.5%, 3%, 2.1% and 4.4% of 57,979 blood donors, respectively.
  2. HBV was predominant among male donors, while the remaining TTIs were predominant among women.
  3. Donors > 50 years had a significantly high prevalence for all TTIs.
  4. Chronic HCV infection was absent in 500 samples tested and OBI was present in 3%.

The authors reported information that may help to control TTIs and identify chronic HCV carriers and possible cases of OBI among blood donors in Angola. However, within the manuscript, there are important shortcomings that must be addressed. 

  • In the title of the manuscript, it is advisable to introduce a reference to pathogen reduction technology (PRT), which is the topic of the Special Issue.
  • In the line 74 of the Introduction Section it can be added that the information obtained will be useful for starting a cost-benefit analysis resulting from the introduction of PRTs.
  • A limitation of the study is represented by the fact that the screening of the parameters studied is not homogeneous because different tests have been used over the years with a sensitivity that has improved in recent years. This could, therefore, have led to an underestimation of the positive findings in the first years considered in the study. A sentence explaining this concept should be included in the discussion.
  • The editing and the formatting of tables need to be improved. In particular:

- Table 1 should be divided into three parts and columns reporting the years should somehow be separated to make them easily legible.

- In Table 2 should be specify that “Total” is referred to the total number of tested donors for each marker. In the title of table please correct the year 2029.  

- Table 3: it is necessary to specify that the number reported refers to the tested donors and for each data section always insert the total number of donors.

Legends should be added to the tables.

  • In paragraph 2.3.1 (line 123) it should be specified that the data refer to the year 2007, as reported in the supplementary file. For clarity it is appropriate to write: “500 tested samples with respect to the total number of samples” (which must be reported). The same should be done for the number of samples reported in line 130 of the paragraph 2.3.2.
  • Line 321. I would like to spend a few more comments on the inactivation of pathogens, with particular regard to the effectiveness of the different methods on the different pathogens for which donors are regularly screened.
  • Line 383. Enter the sample size.
  • Line 405. Considering that PRTs are currently only available for plasma and platelets, it is not appropriate to speak in general of impaired blood quality but of individual blood components.
  • Table Supplement 1. i) Explain that the total donors are not those enrolled in 2007 but those tested for anti-HCV. ii) It is not clear why it is reported that in 2007 500 donors were tested, while in Tables 1 and 2 for 2007 the donors tested for anti-HCV are 848.

Author Response

Manuscript Title: Transfusion transmissible infections in the province of Bié, Angola: a 15-year follow-up

Corresponding Author: Lia Laura Lewis-Ximenez

REVIEWER 2

Comments

#1. Line 74 of the Introduction Section it can be added that the information obtained will be useful for starting a cost-benefit analysis resulting from the introduction of  PRTs.

ANSWER: included sentence in the Introduction Section.

# 2. A sentence explaining the lack of homogenous screening included in the discussion.

ANSWER: The authors have included a few sentences in the Discussion as suggested by the reviewer and an additional sentence in the Discussion  (last paragraph of 3.5) under 3.3 HCV as well.

# 3. The editing and the formatting of the tables need to be improved.

       Table 1 should be divided into three parts and columns.

ANSWER:: Table 1 was reformatted as suggested.

# 4. Table 2 should be specify that ¨ Total¨ is referred to the total number of tested donors for each marker. In the title of table please correct the year 2029

ANSWER: Legend in Table 2 explains Total and the year 2029 was corrected

# 5. Table 3: it is necessary to specify that the number reported refers to the tested donors and for each data section always insert the total number of donors

ANSWER: Included legend to explain and inserted total number of blood donors as suggested

 # 6. Table 4. Legends should be added to the tables

ANSWER: legends added to all tables

 # 7. In paragraph 2.3.1 (line 123) it should be specified that the data refer to the year 2007, as reported in the supplementary file. For clarity it is appropriate to write: “500 tested sample with respect to the total number of samples”  ( which must be reported) # # 8. The same should be done for the number of samples reported in line 130.

ANSWER: Included information in lines 123 and 130 as recommended.

# 9.Line 321. I would like to spend a few more comments on the inactivation of pathogens, …

ANSWER: We have added a few more comments regarding PRT, whole blood, and TTIs in the last few lines of the Discussion. (3.6.2) and also included four new references (56-59).

# 10. Line 383. Enter the sample size

ANSWER: We have included the sample size as recommended

# 1 Line 405. Considering that PRTs are currently only available for plasma and platelets, it is not appropriate to …..

Answer: We have deleted this sentence and substituted it with a different phrase.

#12 Table Supplement 1

  1. Explain that the total donors are not those enrolled in 2007 but those tested for anti-HCV

Answer: Included legend under Table Supplement 1 explaining

  1. Ii) It is not clear why it is reported that in 2007 500 donors were tested, while in Tables 1 and 2 for 2007 the donors tested for anti-HCV were 848. LIA

Answer: Included legend under Table Supplement 1 as NOTE explaining the differences between the 500 blood donor samples tested for anti-HCV in Brazil and the 848 blood donors tested for anti-HCV obtained from secondary data (Hospital Geral do Bié, Angola).

The manuscript had been sent for editing, attached certificate, before submission to Journal. 

Reviewer 3 Report

A retrospective cross-sectional study was conducted at the Hospital Geral do Bié (HGB), in Angola, to analyse around 58000 blood donor data from January 2005 to December 2008, and from January 2013 to December 2020. Results showed high prevalence of TTIs among blood donors in Angola and highlighted a  lower prevalence in voluntary blood donors. Moreover, from frozen samples of blood donors in 2007, the authors investigated chronic HCV carriers (n=500) and possible occult HBV infection (n=170) detecting no chronic HCV and OBI in 3%. The study is well conducted and adds interesting data about Transfusion transmissible infections (TTIs), still scarce in African region.

Some points to be addressed

Lines 71 and 153: Personal communications should not be numbered and added to the references, but only reported in the text.

Footnotes to tables 2-4 should be added.

Discussion section is too long: several sentences can be removed or shortened because they are not relevant to the study results, e.g.: lines 180-183; lines 185-190.

lines 240-241  “Despite the improvement in the testing assays, the prevalence of HIV in 2020 was  comparable to the general prevalence observed over the study period, which supports the high specificity and sensitivity attributed to RDT for HIV [3] “ This sentence meaning is not clear and should be rephrased.

Author Response

Manuscript Title: Transfusion transmissible infections in the province of Bié, Angola: a 15-year follow-up

Corresponding Author: Lia Laura Lewis-Ximenez

REVIEWER 3

# 1. Line 71 and 153: Personal communications should not be numbered and added to the references, but only reported in the text.

Answer: According to the journal instructions, personal communications must also be placed in the reference section. Below is text taken from the "Instructions for Authors" section that shows this statement (marked in bold)

"Unpublished Data

Restrictions on data availability should be noted during submission and in the manuscript. "Data not shown" should be avoided: authors are encouraged to publish all observations related to the submitted manuscript as Supplementary Material. "Unpublished data" intended for publication in a manuscript that is either planned, "in preparation" or "submitted" but not yet accepted, should be cited in the text and a reference should be added in the References section. "Personal Communication" should also be cited in the text and reference added in the References section. (see also the MDPI reference list and citations style guide)."

# 2. Footnotes to tables 2-4 should be added

Answer: Footnotes were added as recommended.

# 3. Discussion section is too long: several sentences can be removed or shortened because they are not relevant to the study results, e.g.: lines 180-183; lines 185-190.

Answer: We have removed lines 183 – 190. We, however, would like to keep lines 180-183 which discuss male predominance in Angola and other SSA countries.

# 4. Lins 240-241 “ Despite the improvement...”. Sentence should be rephrased.

Answer: Sentence rephrased.

The manuscript had been sent for editing, attached certificate, before submission to Journal. 

Round 2

Reviewer 1 Report

Revised manuscript has been partially improved. However, several issues still remain.

  • What is the table (?) between Table 1 and 2?
  • Conclusion of abstract: Will the authors use infectious blood products? In present technology, we cannot economically remove several pathogens.
  • Figure 3 and 4 are still unclear. Authors should remake these figure.

Author Response

# 1  What is the table (?) between Table 1 and 2?

  • ANSWER: There is no table between table 1 and 2. We forgot to delete the previous table completely, have excluded.

# 2 Conclusion of abstract: Will the authors use infectious blood products? In present technology, we cannot economically remove several pathogens.

  • No we will not use infectious blood products as we are aware that there is no available commercial PRT for whole blood at the moment. However, once accessible, low-income countries, especially those with high prevalences for infectious diseases could benefit from this technology as suggested by a few authors such as S. Amar El Dusouqui et al (2019) and J.P. Allain (2017). We added a few words to the sentence in the conclusion of the abstract that clearly specifies the unavailability of the PRT for whole blood.

# 3 Figure 3 and 4 are still unclear. Authors should remake these figures.

  • ANSWER: Figures 3 and 4 were substituted as suggested and legends adjusted accordingly. 

# 4 English language and style are fine/minor spell check required

  • ANSWER:  We have reviewed the manuscript and corrected a few misspelled words, corrected spacing, and improved the English whenever required. 

Reviewer 2 Report

Although some of the requests have not been met (e.g. the title change), the manuscript has nevertheless been improved enough.

Author Response

In the title of the manuscript, it is advisable to introduce a reference to pathogen reduction technology (PRT), which is the topic of the Special Issue.

  • ANSWER: Sorry for skipping this suggestion and have added PRT as recommended

Transfusion transmissible infections in blood donors in the province of Bié, Angola, during a 15-year follow-up, imply the need for pathogen reduction technologies

Reviewer 3 Report

The paper has been improved by revisions.

Author Response

#1 English language and style are fine/minor spell check required

  • ANSWER:  We have reviewed the manuscript and corrected a few misspelled words, corrected spacing, and improved the English whenever required.